# The Extraction and Impact of Essential Oils on Bioactive Films and Food Preservation, with Emphasis on Antioxidant and Antibacterial Activities—A Review

**DOI:** 10.3390/foods12224169

**Published:** 2023-11-18

**Authors:** Sohail Khan, Abdullah A. A. Abdo, Ying Shu, Zhisheng Zhang, Tieqiang Liang

**Affiliations:** 1College of Food Science and Technology, Hebei Agricultural University, Lekai South Avenue, Baoding 071000, China; sohailkhan6566@yahoo.com (S.K.); drabdoabdullah2021@gmail.com (A.A.A.A.); hebaushuying@hebau.edu.cn (Y.S.); 2Department of Food Science and Technology, Faculty of Agriculture and Food Science, Ibb University, Ibb 70270, Yemen; 3Hebei Layer Industry Technology Research Institute, Economic Development Zone, Handan 545000, China

**Keywords:** extraction, essential oil, bioactive films, bioactivity, food application

## Abstract

Essential oils, consisting of volatile compounds, are derived from various plant parts and possess antibacterial and antioxidant properties. Certain essential oils are utilized for medicinal purposes and can serve as natural preservatives in food products, replacing synthetic ones. This review describes how essential oils can promote the performance of bioactive films and preserve food through their antioxidant and antibacterial properties. Further, this article emphasizes the antibacterial efficacy of essential oil composite films for food preservation and analyzes their manufacturing processes. These films could be an attractive delivery strategy for improving phenolic stability in foods and the shelf-life of consumable food items. Moreover, this article presents an overview of current knowledge of the extraction of essential oils, their effects on bioactive films and food preservation, as well as the benefits and drawbacks of using them to preserve food products.

## 1. Introduction

Essential oils are significant secondary metabolites found in the flowers, fruits, leaves, buds, and stems of aromatic plants [1], accounting around 5% of dry matter [2], and are highly valuable for commercial applications [3]. A number of factors, for instance plant part, geographical origin, time, season of harvest season, extraction method, and solvent used for extraction, influence their yield and chemical composition. They are quite famous for their biological activities, for instance anti-inflammatory, anti-cancer, antimicrobial, and antioxidant properties [4], which promotes their utilization in the pharmaceutical, cosmetic, chemical, and food industries [5]. According to researchers, essential oils have been extensively incorporated in biodegradable films as an additive due to their antibacterial [6] and antioxidant activities for food preservation [7]. Biopolymers like polysaccharides, proteins, and lipids could be used alone or in combination to develop edible films with better technological characteristics and desirable properties [8]. Biofilms offer numerous benefits, such as eco-friendliness, biocompatibility [9], better barrier properties, cost-effectiveness, and being carriers of food additives like vitamins, antioxidants, and antimicrobial agents [10]. Regarding consumer demand for safe and healthy products, researchers have developed active packaging by including antimicrobials and antioxidant agents in polymeric matrices to enhance product quality and nutritional value [11]. Active packaging increases the quality, safety, and stability of food for a long time by preventing the oxidation and pathogenic microorganism activity that contaminate food [12]. Biomaterials allow for the incorporation of antioxidant and antimicrobial materials [13], like essential oils for active film development [14]. Among plant-based active ingredients, essential oils (EOs) have received significant attention from food researchers related to their safety, unique flavors and aromas, and antioxidant and antimicrobial contents to prepare “bioactive packaging” for increasing the safety and shelf-life of food [15] and to assist in reducing hydrophilic behavior [16]. In past years, EOs such as clove (*Syzygiumaromaticum*), oregano (*Origanum vulgare*), cinnamon (*Cinnamomum zeylanicum*), peppermint (*Mentha piperita*), anise (*Pimpinella anisum*), lemongrass (*Cymbopogon citratus*), ginger (*Zingiber officinale* Roscoe), thyme (*Thymus vulgaris*), rosemary (*Salvia rosmarinus*), and other plants have been used for food preservation [17]. Due to their strong antimicrobial activity against various pathogens and antioxidant activity [18], a number of researchers have recommended their use in food formulation, preservation, and packaging [19,20]. Despite the beneficial potential of essential oils in active packaging, they are volatile, insoluble in water, and produce certain smells that are not appropriate in some applications [21]. The direct incorporation of EOs in food is not effective since their antimicrobial and volatile components could be partially inactivated when exposed to light, oxygen, and high processing temperatures. In addition, the irregular discharge of these compounds may have negative antibacterial and sensory effects. To overcome these problems, the incorporation of these compounds into coatings [22] or packaging films can help to manage their release during the product’s shelf-life [15]. So far, only a few comprehensive reviews on the bioactivities of different EOs with limited scope have been published. But recently, three reviews on the application of EOs in the development of nanocarriers as active food packaging were published by Doost, Nasrabadi [15], Jugreet, Suroowan [23], and Rehman, Jafari [24]. Thus, here we discuss the extraction of essential oils, their utilization in bioactive film development, their bioactive properties, and applications in the food industry to enhance the shelf-life of products by preventing oxidation and microbial growth.

## 2. Extraction of Essential Oils

Essential oils are obtained from different plant parts using different extraction techniques depending on the material and state. Furthermore, incorrect extraction techniques can affect the natural qualities and bioactivity of essential oils, making extraction methods important for quality control [25] since they can lead to the loss of natural characteristics and bioactivity [26]. More importantly, physical changes such as odor, discoloration, and viscosity can occur in extreme cases when using improper extraction techniques [27]. These unwanted changes in essential oils should be avoided. There are several methods for the extraction of essential oils, some of which are discussed below.

### 2.1. Distillation

A typical method to separate a combination of fluids into discrete fractions is the method of distillation [28]. This process uses the equilibration diffusion phenomenon in gas–liquid separation procedures, just like adsorption [29]. Further, this method relies on the range of boiling points or the relative volatility of the mixture’s constituent parts. Similarly, distillation separates fluid mixtures based on their boiling points, with more volatile components vaporizing first and less volatile components remaining in the liquid phase [30]. Although distillation requires high energy, its numerous benefits, such as efficiency, transfer rates, and flexibility, make it popular in the market and industries [31].

Recently, bioactive compound separation, such as essential oils, through distillation has gained significant technological interest. A variety of distillation procedures including steam distillation, hydrodistillation, and hydrodiffusion used for essential oil separation from biomass are discussed below.

#### 2.1.1. Steam Distillation Method

Steam distillation is an extraction practice that is suitable for the separation of aromatic compounds of essential oils from temperature-sensitive plants [32]. The oil-containing materials placed in the distillation equipment without maceration are exposed to steam. The injected steam break opens the pores and releases the oil content by passing through the raw biomaterial. The system produces a mixture of vapor and raw essential oil, which are condensed, and the vapor is eliminated to obtain the essential oil [33]. Depending on the plant material and oil extraction difficulties, the procedure can be performed under various temperatures but it should be high enough to break down the cell structure to release the essential oil or aromatic compounds [34]. A new procedure was developed to enhance oil yield and decrease polar molecule loss in wastewater during steam distillation. In this system, the plant material sits above the steam source, as shown in Figure 1, so only the steam passes through it. This process requires a minimum amount of steam, which results in less water content in the distillate. Moreover, the dissolution of water-soluble compounds in the aqueous fraction of the condensate is also reduced [35]. Steam distillation extracts 93% of essential oils, but other techniques can be used to extract the remaining 7%. Since the operating temperature hardly goes over 100 °C, it has the advantage of inducing low thermal oil degradation, as reported by Yildirim, Cakir [36]. Also, the amount and the quantity of heat can be efficiently controlled. Apart from the useful aspects of technique, it also has a few disadvantages. Uniform material distribution, size, and packing are crucial in distillation to avoid steam blockage; moreover, the capital investment required for this process is large and it is very expensive for low-price products.

#### 2.1.2. Hydrodistillation Method

This traditional method is used to extract essential oils from plant materials like flowers or wood, but mostly water-insoluble natural products having high boiling points are isolated through this process [37]. This process can be performed prior to dehydration of plant materials without the use of organic solvent. The plant materials are boiled after being submerged entirely in water, as shown in Figure 2. The extracted oils are protected from overheating to a certain extent by the surrounding water. Steam and hot water are the most important variables in the release of bioactive chemicals from plant tissue. Oil and bioactive substances automatically separate from water as the water and oil mixture pass from the condenser to the separator [38]. Okoh, Sadimenko [39] examined the properties and yield of rosemary (*Rosmarinus officinalis* L.) essential oil extracted through hydrodistillation (HD) and solvent-free microwave extraction (SFME). Both SFME and HD produced volatile fractions with total yields of 0.31%, and 0.39%, respectively. SFME-extracted oil had a higher proportion of oxygenated monoterpenes (28.6%) than HD oil (26.98%), while HD oil had a higher proportion of monoterpene hydrocarbons (32.95%) than SFME-extracted oil (25.77%). An advanced HD technique called microwave-assisted HD (MAHD) was studied by Golmakani and Rezaei [40] and was considered superior in terms of consuming less processing time (75 min compared to 4 h in HD) and energy. Another advanced HD technique known Ohmic-assisted HD (OAHD) took 24.75 min to extract essential oil from thyme, with some differences in the essential oil compound extracted by HD [41]. However, some volatile components could be lost by the use of a high extraction temperature. This downside limits its usage for thermo-labile compound extraction [42]. A large amount of mother liquor having water-soluble phenolics may be produced by this process. This problem can be solved by treating the remaining wet residue (herb and water) with hydrodistillation to take full advantage of extraction waste in order to obtain phenolic acids [43].

#### 2.1.3. Hydrodiffusion

It is a kind of steam distillation, which is different from other forms of steam distillation, only owing to the steam inlet path to the container. This process is applied to extract essential oils from dry materials, which are not affected by boiling temperature [44]. In the steam distillation method, the steam is applied from the bottom, whereas it is applied from the top during hydrodiffusion. Some innovations were carried out to improve its performance in the extraction of essential oils. For example, HD and innovative microwave hydrodiffusion (MHG) methods were compared in the extraction of essential oil from rosemary leaves (*R. officinalis*) [45]. The MHG method outperformed traditional practices regarding limited extraction time (15 min compared to 3 h by HD), environmental impact, cleaner features, and improved antimicrobial and antioxidant activities. Another innovative process, called microwave steam diffusion (MSDf), was studied in [46] to extract essential oils from orange peel by-products. Similar aromatic component profiles were seen in the extracted essential oils using MSDf for 12 min and advanced steam diffusion (SDf) for 40 min. The advantage of this procedure is that it can be carried out under vacuum or low pressure to reduce the steam temperature below 100 °C. Moreover, this process is considered better than steam distillation due to higher oil yield and short processing time with less steam usage [47].

### 2.2. Solvent Extraction

This technique has been used for polyphenol extraction from delicate or fragile plant parts that cannot tolerate the high temperature of steam distillation. A variety of solvents, including petroleum ether, acetone, hexane, ethanol, or methanol, can be used for extraction [48]. The effectiveness of organic solvents depends on their selectivity towards target compounds/chemicals, which in turn depends on the molecular structure of that compound or chemical [49].

This method involves the mixing of plant material with the solvent, followed by heating to extract the essential oil, and then the dissolution and distribution of soluble organic compounds throughout the solvent. The filtrate of the solvent and essential oil mixture is concentrated by solvent evaporation. The concentrate consists of a concrete (a mixture of wax, essential oil, and fragrance) or resin (resinoid). The oil is then extracted by distilling the concentrate at a low temperature after being combined with pure alcohol. The alcohol absorbs the fragrance and an absolute aromatic oil remains after evaporating the alcohol. This process is relatively time-consuming, which makes the oil more expensive than oils extracted by other methods [50]. Ozen, Demirtas [51] investigated the chemical composition and antioxidant activity of *Thymus praecox subsp. skorpilii var. skorpilii* (TPS) essential oils extracted with various solvents. The two main ingredients of TPS essential oil are thymol (40.31%) and o-cymene (13.66%). The ethanol, methanol, and water extracts showed outstanding free radical scavenging activity. The highest levels of total phenolics (6.211 mg gallic acid/g dry weight) and flavonoids (0.809 mg quercetin/g dry weight) were found in the water extract [52]. Furthermore, according to Oreopoulou, Tsimogiannis [53], the highest antioxidant activity was present in the water extract compared to the other extracts (ethyl acetate, hexane, methanol, and dichloromethane). However, the final product had a solvent residue which could cause toxicity, allergies, and alter the immune system [54]. This method has some significant shortcomings, particularly the lengthy extraction time and usage of huge amounts of organic solvents. Yet the method is still employed and studied to improve the terms and conditions for industrial scale applications [55].

### 2.3. Supercritical Fluid Extraction

Supercritical fluid extraction (SFE) is a modern technique used to extract heat-sensitive substances. It is a better alternative to traditional extraction methods like solvent extraction and steam distillation due to its shorter operation time, less organic solvent consumption, and production of clean extracts [56]. The most common supercritical fluid used in SFE is carbon dioxide due to its modest critical conditions [26]. Carbon dioxide converts into liquid at high-pressure conditions, which is a very safe and inert medium for aromatic molecule extraction from raw materials. The final product does not have a solvent residue because the liquid CO_2_ reverts into gas under normal pressure and temperature [57]. Supercritical fluid extraction of targeted compounds similar to conventional extraction procedures is dependent on different factors, including temperature, pretreatment of materials, particle size, time, pressure, solvent-to-feed ratio, and solvent flow rate [58]. These perimeters affect the extraction efficiency in terms of yield and targeted component recovery. Despite the considerable solubility of essential oils in supercritical CO_2_, the extraction rates with pure CO_2_ are relatively slow [59]. High recoveries are produced when the 15 min static extraction approach with methylene chloride as a modifier is combined with the 15 min dynamic extraction method utilizing pure CO_2_ [60]. Furthermore, supercritical fluid extraction can be used to gather volatile compounds like monoterpenes to a level of >90%. It can recover some organic compounds which cannot be extracted through traditional techniques like hydrodistillation. Karrar, Sheth [61] presented that SFE is more economical than steam distillation due to the higher energy consumption and lower yield of steam distillation.

## 3. Biomaterials

Active films are often made from single polymers like polysaccharides, proteins, lipids, or a mixture of these polymers [62]. The film-forming matrix is an important factor influencing film properties. These properties are usually assessed through important parameters such as mechanical strength, water vapor permeability (WVP), and oxygen permeability (OP). Compared to hydrophobic materials, water-soluble biomaterials such as proteins and polysaccharides mostly exhibit better mechanical properties and higher barrier capacities against O_2_ and CO_2_. However, due to their hydrophilic nature, these biomaterials exhibit higher water permeability [63] and are not compatible with some food processing conditions like high pressure. Waxes and other lipids cannot form films, in contrast to other biomaterials [64]. This shortcoming can be completely overcome by mixing different biopolymers, enzymatic modification of polymers, and addition of hydrophobic materials (e.g., waxes and oils) [65]. Due to their low polarity, beeswax and other lipid structure materials can be combined with water-soluble polymers to develop high moisture barrier films [66]. Recently, the protective qualities of edible films have been improved with the addition of various components, like nanoparticles. For example, the incorporation of chitin nanofibrils into carrageenan film improved its mechanical properties and reduced its water vapor permeability [67]. Another study demonstrated that the incorporation of essential oil increased the hydrophobicity of the film to prevent moisture transfer [68]. Essential oils also show antimicrobial, antioxidant, and anti-inflammatory properties, which make them suitable for incorporation with biomaterials to produce bioactive films. Figure 3 shows natural polymers used to prepare biodegradable films for food preservation.

### 3.1. Polysaccharide-Based Films Loaded with Essential Oils

Polysaccharides are natural polymers used for film development due to their outstanding mechanical and structural qualities. Polysaccharide-based films have better oxygen barrier capacity due to their organized hydrogen bond network [63]. However, due to their hydrophilic structure domain, they do not behave well in moisture resistance. The main polysaccharides used as biopolymers in edible film preparation are starch cellulose, pectin, carrageenan, alginates, agar, chitosan, and pullulan [69]. These polysaccharides have better film-forming properties, and the prepared films are generally tasteless, odorless, flexible, transparent, and provide a great barrier against O_2_ and CO_2_ [70]. Although they are not an effective barrier against water vapor due to their high hydrophilicity, this problem can be solved by incorporating a bioactive material like an essential oil into a film medium, increasing its hydrophobicity to reduce the WVP and enhancing the antibacterial and antioxidant activities of the film [71]. For instance, the antibacterial activity of carboxymethyl cellulose (CMC) and polyvinyl alcohol (PVOH) film was enhanced through clove essential oil. The film lowered the total viable counts of packed meat and preserved it for a longer time (12 days) compared to the control film (4 days) under refrigerated conditions [72]. Similarly, the antioxidant and antibacterial activities of chitosan and carboxymethyl cellulose-based composite film were improved with the integration of glutaraldehyde and cinnamon essential oil [73]. Another interesting outcome was reported by Sarıcaoglu and Turhan [74], who used Bombacaceae gum with cinnamon leaf essential oil in a bioactive film. The results revealed that essential oil incorporation in Bombacaceae gum film improved its hydrophobicity, flexibility, and antioxidant and antimicrobial properties. Moreover, it inhibited the lipid oxidation of salmon fillets stored at 4 °C. Mahcene, Khelil [75] stated that the incorporation of *A. herba alba Asso*, *R. officinalis* L., *M. pulegium* L., and *O. basilicum* L. essential oils to sodium alginate-based film decreased the WVP and OP and increased the antioxidant and antimicrobial activities against pathogenic bacteria. The film developed by de Oliveira Filho, de Deus [76] from chitosan presented a lower WVP and improved antibacterial and optical properties with the incorporation of *Citrus limonia* essential oil. Esterified tapioca starch film supplemented with oregano essential oil presented a reduction in the WVP and improved mechanical, antioxidant, and antifungal properties [77]. Similarly, ginger essential oil improved the antimicrobial properties of chitosan-based film against both Gram-positive and Gram-negative bacteria, like *Streptococcus* spp., *Staphylococcus aureus*, *Bacillus subtilis*, *Escherichia coli*, *Salmonella* spp., and *Pseudomonas aeruginosa*, resulting in shelf-life extension of food products [78]. There are many more reports presenting improvements in the preservation properties of films by incorporating essential oils.

### 3.2. Protein-Based Films Loaded with Essential Oils

Proteins have also received huge attention both for film and biomaterial production [79]. Proteins are heteropolymers composed of more than one type of monomer and contain extensive types of functional groups [80]. Proteins play a vital role in food packaging materials because of their availability, transparency, film-forming capacity, and better barrier properties against CO_2_ and O_2_ [64]. The properties of protein-based films are determined by the length of the protein chain and the amino acid sequence. Some materials, like whey protein, gelatin, wheat gluten, casein, soy, collagen, keratin, corn zein, and egg albumen, are commonly used for film preparation [81]. Generally, for film formation, the proteins are denatured by acid, base, heat, and solvent to develop the required elongated and cohesive structure to generate a barrier against carbon dioxide and oxygen [69]. However, protein films are less resistant to both water diffusion and mechanical stress and might become a little brittle on drying [82]. Due to the heteropolymeric nature of proteins, it is possible to modify their characteristics through chemical, enzymatic, or physical means to give them the desired characteristics needed for food packaging [83]. Proteins serve as good carriers for a variety of bioactive substances, including vitamins, minerals, phenolic compounds, peptides, and essential oils [84]. In this regard, bioactive compounds have been added to packaging materials to maintain the quality of food and protect it from environmental stresses [85]. Among bioactive materials, essential oils are often used against a number of pathogenic microorganisms due to their better antioxidant and antimicrobial effects [71]. Many EOs, for example cinnamon (*Cinnamomum verum*), rosemary (*Salvia rosmarinus*), clove (*Syzygiumaromaticum*), garlic (*Allium sativum*), mustard (*Brassica nigra*), forsythia (*Forsythia suspensa*), oregano (*Origanum vulgare*), lemon grass (*Cymbopogon citratus*), and thyme (*Thymus vulgaris*), have been used to increase the antibacterial, antioxidant and barrier properties of films [86]. Lemon and bergamot EOs were integrated into whey protein isolate-based film, which exhibited a remarkable response against microorganisms, water vapor, and oxygen permeability [87]. Sarıcaoglu and Turhan [74] stated that the incorporation of clove (*Syzygium aromaticum* L.), thyme (*Thymus vulgaris* L.), and rosemary (*Rosmarinus officinalis* L.) essential oils into chicken meat protein film reduced the WVP and improved antioxidant and antibacterial activities [74]. Similar effects were determined in other reports by the incorporation of rosemary, mint [57], and clove essential oils [88]. Another whey protein (WP)-based film supplemented with Thymbra (*Satureja capitata*, L.) essential oil presented an improvement in antimicrobial activity, indicating its potential to be utilized as a preservative material in food packaging [88]. Moreover, film prepared with Nile tilapia (*Oreochromis niloticus*) protein isolate (NTPI) having clove and oregano essential oils presented a lower water vapor permeability and better mechanical and antibacterial properties [89]. Recently, Seydim, Sarikus-Tutal [90] reported the impact of WPI-based edible films containing garlic and oregano EOs on the retardation of microbial growth in sliced Kashar cheese. The WPI films supplemented with EOs provided great microbial stability against *Listeria Staphylococcus aureus, monocytogenes*, *Escherichia coli* O157:H7, *Penicillium* spp., and *Salmonella enteritidis* during storage. Many other essential oils have been used to improve the functional properties, like WVP, and the antioxidant and antimicrobial properties of protein-based films.

### 3.3. Composite Films Loaded with Essential Oils

Composite films are prepared for their better functional properties. This can be performed to obtain complementary benefits from each component. For instance, hydroxypropyl starch/zein bilayer edible film presented a lower WVP than monolayer starch film [91]. Protein and starch-based materials are commonly used for composite edible film production [92]. The film morphology can be altered by combining two or more polymers with some active fillers to obtain better controlled-release qualities.

Thus, smart blending can be used to create new packaging materials to control the release of active materials from the films [65]. Therefore, the incorporation of active agents is helpful to promote the preservation properties of films. Due to the increasing interest in essential oils in the scientific community in recent years regarding food product safety, low toxicity, and significant antioxidant and antimicrobial activities, it has been considered suitable to incorporate EOs with other materials for the development of composite films [93]. Numerous studies have demonstrated that in combination with EOs, biodegradable films and coatings are very effective in enhancing food shelf-life. For example, chitosan-based film having ginger essential oil and montmorillonite was used for beef packaging. Due to its commendable antioxidant and antibacterial activities, the film was very effective in extending the shelf-life of beef [94].

Another report showed that the incorporation of lemon EO into grass carp collagen-chitosan film preserved pork for 21 days by improving its barrier, mechanical, antibacterial, and antioxidant properties, as reported by Jiang, Lan [95]. Another researcher developed zein/pullulan-based biodegradable bilayer film with added licorice essential oil. The film showed good mechanical, optical, and barrier properties and retarded lipid oxidation and microbial growth [96]. Similarly, the addition of orange peel essential oil (OPEO) in fish skin gelatin and chitosan-based film promoted its light and water vapor barrier properties and improved its antioxidant and antibacterial activities compared to the control film [97]. Apple peel pectin/potato starch-based composite film having ZrO_2_ nanoparticles and *Zataria multiflora* essential oil was prepared for quail meat packaging. The film having essential oils and ZrO_2_ nanoparticles showed great improvements in antioxidant, antibacterial, and mechanical properties and increased the shelf-life of quail meat [98]. The results of other composite films prepared with chitosan-carboxymethyl cellulose and added cinnamon essential oil (CEO), oleic acid (OA), and glutaraldehyde (GL) showed significant improvements in antioxidant, antimicrobial, mechanical, and physical properties [73]. Different food products have been preserved by combining various essential oils with materials in film form.

## 4. Bioactivity of Essential Oil-Added Film

The most common bioactivities found in essential oil-added films are antioxidant and antimicrobial activities.

### 4.1. Antimicrobial Activity

Foodborne diseases cause devastating economic losses and threaten consumers’ lives. In this regard, it was found that bioactive films having essential oils (EOs) would be the best choice to develop smart food packaging [99]. Generally, the supplementation of EOs into films enhances their antibacterial activities against foodborne pathogens.

Numerous reports have stated that films having EOs displayed strong antimicrobial activity. The antibacterial activity of chitosan film against *Staphylococcus aureus* was improved with the addition of *Citrus limonia* essential oil (CLEO) [76]. In another report, chitosan film having thyme oil had more powerful antibacterial activity against *E. coli* spp. [99]. Furthermore, the antibacterial activity of chitosan films against both *E. coli* spp. and *Bacillus subtilis* were enhanced with the incorporation of ginger oil [100] and clove essential oil (CEO) [76]. Similarly, CEO enhanced the antibacterial effect of a composite edible film against some targeted microorganisms like *E. coli* and *L. innocua* [101].

The antimicrobial property of active agar (AG) bilayer film was increased with the addition of neem essential oil (NEO) by reducing all tested bacteria strains, such as *S. aureus* and *E. coli* [102]. In line with the study, 3% oregano essential oil (OEO) added to starch film displayed significant antibacterial activity against *Escherichia coli*, *Staphylococcus aureus*, and *Bacillus subtilis* [103]. Chitosan/fish skin gelatin film containing orange peel essential oil (OPEO) exhibited higher antimicrobial effects against *Staphylococcus aureus*, *Escherichia coli*, *B. subtilis*, *C. albicans*, and *P. aeruginosa* [97].

Moreover, the antimicrobial activity of rosemary EO encapsulated in carboxyl methylcellulose film against *lactic acid bacteria* and *Pseudomonas* spp. was proven by Moeini, Pedram [63].

Antifungal activity was also reported. The addition of turmeric essential oil in chitosan film significantly boosted its inhibitory ability against *conidial* formation and *Aspergillus flavus* growth [101]. Similarly, OEO-added film presented good antifungal activity [77]. Thus, the patent promotion of the antimicrobial activity of the bioactive film could be ascribed to the essential oil addition.

The antimicrobial activity could be due to the presence of bioactive compounds in EOs such as rosemanol, epirosmanol, and rosmarinic acid in ginger oil (GO), limonene in clove essential oil, and sterols, active ester derivatives, and triterpenoids in neem essential oil (NEO), which act as antimicrobial agents [77,104,105]. However, the compound responsible for this activity in some essential oils is not clear. In most cases, the antimicrobial mechanism of bioactive compounds in EOs is by damaging cell membranes through the interaction between essential oils and membrane proteins [104]. Moreover, the cytoplasmic membrane of the bacterium is affected by the aromatic and phenolic components of the oils, which alters its activity and disrupts enzymatic systems [100]. However, the antibacterial mechanisms of some EOs need more investigation.

The effect of EOs on bacteria varies according to bacteria type; most EOs are more effective against Gram-positive bacteria than Gram-negative species. For example, the antibacterial activity of pullulan-based film having rockrose (*Cistus ladanifer*) essential oil (REO) against Gram-positive bacteria was higher than that against Gram-negative bacteria [106]. Similarly, chitosan film with added *Citrus limonia* essential oil (CLEO) did not present any antibacterial effect against *Escherichia coli* (Gram-negative bacterium) but its antibacterial activity was observed against *S. aureus* (Gram-positive bacteria) [105]. It was found that Gram-positive bacteria have a cell envelope that is functionally and structurally less complex than that of Gram-negative bacteria, which have molecules on their membranes that act as a barrier to decrease hydrophobic compound formation. Furthermore, lipopolysaccharide molecules on Gram-negative bacteria membranes act as a barrier, decreasing hydrophobic compound formation [105]. Furthermore, the antimicrobial activity of EOs differs according to EO type. It was found that clove essential oil (CEO) in chitosan (CS) film showed the greatest inhibition against microorganisms, compared to melaleuca essential oil [107]. Chitosan film supplemented with tea tree oil presented better antibacterial activity against *S. aureus* and *C. albicans* [108]. The data regarding EO comparisons are limited. Similarly, the antimicrobial activity of EOs depends on their concentration in films, since it is positively correlated with EO level [97]. Further investigation is needed to evaluate the correlation between antimicrobial activity and EO concentration.

In contrast to the results mentioned above, adding caraway essential oil (1% *v*/*v*) did not exhibit any antimicrobial effect in chitosan film [101]. *Citrus limonia* essential oil (CLEO) added to chitosan film also did not present any antibacterial activity against *Escherichia coli* [105]. These results suggest that antimicrobial activity is strongly affected by EO type.

### 4.2. Antioxidant Activity

Another biological property of EOs is antioxidant activity, which protects food from oxidative damage [109]. Chitosan film incorporating ginger essential oil (GEO) showed high scavenging capacity for superoxide and hydroxyl radicals. The antioxidant property of the film was increased by mango seed oil and *Mentha spicata* essential oil, compared with film without essential oils [100]. In one study, rosemary essential oil improved the antioxidant activity of gelatin and chitosan/gelatin film [104]. Furthermore, the inhibitory activity of fish skin gelatin (FSG)/soluble chitosan (CH) film against free DPPH and ABTS radicals was promoted with the addition of orange peel essential oil [97]. Similarly, chitosan film supplemented with *Berberis crataegina* seed oil showed enhanced antioxidant activity [102]. Clove essential oil (CEO) increased the antioxidant activity of gelatin/myofibrillar protein film (Gel-Sur film) [110]. Different concentrations of ascorbic acid (0, 1, 2%) in the gelatin layer and *Hyssopus officinalis* essential oil (0, 0.75, 1.5%) in the frankincense layer of gelatin and frankincense gum film promoted its antioxidant activity [80]. Clove essential oil (EO) also improved the antioxidant activity of Gel-Sur film [110]. Film incorporated with angelica essential oil (AEO) presented an increase in DPPH (2,2-diphenyl-1-picrylhydrazyl) scavenging activity [111]. The antioxidant activity of soluble soybean polysaccharide (SSPS) film was developed by adding cinnamon essential oil nanoemulsion (CNO) [112]. In line with the study, oregano essential oil (OEO) could increase the antioxidant activity of starch film [103]. Oxidized esterified tapioca starch (OETS)-based film containing 0.9% OEO had the greatest antioxidant ability (DPPH, ABTS, and FRAP) compared to film without OEO [77]. The same result was noted by adding OEO to seed gum-based film [113]. Sodium caseinate film containing *Melissa officinalis essential oil* (MOEO) at levels of 0, 5, and 10% *w*/*v* presented high antioxidant activities [114]. The antioxidant activity of agar (AG) film was improved by neem essential oil (NEO) addition [102]. The antioxidant activity of bioactive pullulan-based film was also developed by adding rockrose essential oil (REO) [106].

The antioxidant activity of EOs against radicals could be due to antioxidant ingredients such as phenolic compounds [97]. The active groups in EOs terminate the chain reaction of free radicals by reacting as hydrogen donors with free radicals [111]. For example, the 9% Gel-Sur film containing clove essential oil showed a high ABTS activity and DPPH value on account of phenolic compounds in CEO [110]. Furthermore, the antioxidant ability of the film might be related to the redox properties of effective antioxidants (phenolic acids and terpenoids) of OEO loaded in the film [115]. The antioxidant activity of pullulan-based film with added REO may be due to the main abundant compounds of the mentioned oil (camphene, bornyl acetate, and trans-pinocarveol-pinene) [106]. Many other active compounds, such as arcurcumin, zingiberene, borneol, and camphene, in GEO contributed to improving the scavenging ability against free radicals in film [100]. Moreover, the polyphenols (rosemanol, epirosmanol, and rosmarinic acid) in rosemary essential oil contributed to an increase in the antioxidant activity of gelatin film [104]. Finally, it was found that there was a linear correlation between total phenol content and the radical scavenging ability of EOs [80].

The antioxidant activity of bioactive films incorporating EOs is affected by the EO concentration. For example, the scavenging ability of chitosan film with added ginger essential oil (GEO) against free radicals was improved with an increase in GEO concentration [100]. Similarly, the scavenging ability of FSG/CH film against DPPH and ABTS radicals increased markedly with the addition of orange (*Citrus sinensis* L.) peel essential oil (OPEO) in a concentration-dependent manner [97]. In line with the study, the antioxidant activity (DPPH) significantly increased as a function of *Hyssopus officinalis* essential oil (HO) concentration [80].

In addition, the incorporation of OEO into oxidized esterified tapioca starch (OETS) film enhanced its DPPH radical scavenging ability from 35.21% to 76.98% with an increase in concentration from 0.3% to 1.5% (Lu et al., 2021). The addition of 1.5% of OEO could develop a highly active OETS-based film against free radicals [77]. Similarly, the antioxidant activity of gelatin/chitosan film increased with increasing lemon EO concentration [84]. The antioxidant and antibacterial properties of some other EOs incorporated in bioactive films are given in Table 1.

Taken together, the antibacterial and antioxidant properties of EOs employed in bioactive films have a positive impact on film quality by preventing the growth of pathogenic microorganisms and scavenging free radicals. Thus, films enriched with EOs could be used as effective antioxidant and antimicrobial packaging materials. However, the safety of films with EOs needs further investigation to optimize EO levels, thus enabling their application in foodstuffs without compromising their nutritional characteristics and sensory properties.

**Table 1 foods-12-04169-t001:** Common essential oils incorporated into films and their antimicrobial and antioxidant effects.

Essential Oil	Major Active Components	Film Type	EO Levels in the Film	Findings	Reference
Oregano EO	Phenolic acids and terpenoids	Oxidized esterified tapioca starch film	0.3, 0.6, 0.9, 1.0, and 1.5%	- Improved antifungal activity (*Curvularia Lunata*). - Improved the antioxidant activity against DPPH, ABTS, and FRAP.	[77]
Oregano EO	Phenolic acids and terpenoids	Starch film (*Dioscorea zingiberensis* starch)	1, 2, and 3%	- Improved the antimicrobial activity against *Bacillus subtilis*, *Escherichia coli*, and *Staphylococcus aureus*. - Improved the antioxidant activity against ABTS and DPPH.	[103]
Oregano EO	Phenolic acids and terpenoids	Poly (lactic acid)/poly (trimenthylene carbonate)	3, 6, 9, and 12%	- Improved the antimicrobial activity against *E. coli* and *L. monocytogenes*.	[115]
Licorice EO	Isopropyl palmitate	Zein/pullulan	30%	- Improved the antimicrobial activity against *Enterococcus faecalis* and *Listeria monocytogenes.* - Improved lipid peroxidation inhibition.	[116]
Rockrose EO	Camphene, bornyl acetate, trans pinocarveol, and α-pinene	Pullulan	15%	- Improved the antimicrobial activity against *L. monocytogenes LMG* 16779, *E. faecalis ATCC29212*, *B. cereus ATCC 11778*, and *P. aeruginosa ATCC 27853*. - Improved the antioxidant activity against DPPH radicals and lipid peroxidation.	[106]
Clove EO	Eugenol, α- karyophylene, β-karyophylene, and α-humulene	Edible film	5%	- Improved the antimicrobial activity against *E. coli*. and *L. innocua*.	[101]
Citrus limonia EO	Limonene, nerol, and 1,8-cineole	Chitosan	0.4, 0.8, 1.5, and 2%	- Improved the antimicrobial activity against *Staphylococcus aureus.*	[76]
Lemon EO	Phenolic compounds	Gelatin/Chitosan	0.25, 0.50, 0.75, and 1.0%	- Improved the antioxidant activity against DPPH and ABTS.	[117]
Orange peel EO	Phenolic compounds	Chitosan and fish skin gelatin	0.25, 0.5, and 1.0%	- Improved the antimicrobial activity against *S. aureus*, *B. subtilis*, *E. coli*, *P. aeruginosa,* and *C. albicans*. - Improved the antioxidant activity against DPPH and ABTS.	[97]
Neem EO	Sterols, triterpenoids, and active ester derivatives	Active agar (AG) bilayer film	2.0 g NEO/100 g AG	- Improved the antimicrobial activity against *E. coli* and *S. aureus*. - Improved antioxidant activity.	[102]
Thyme EO	Sterols, triterpenoids, and active ester derivatives	Chitosan	1.0%	- Improved the antimicrobial activity against *Bacillus subtilis* and *E. coli* spp.	[99]
Ginger EO	Eucalyptol (19.36%), (-)-camphene (15.07%), β-bisabolene (11.52%), zingiberene (9.58%), and cineol	Chitosan	0, 0.1, 0.2, and 0.3 (%)	- Improved the antimicrobial activity against *E. coli* spp. and *Bacillus subtilis*.	[100]
Rosemary EO	Polyphenols such as rosemanol, epirosmanol, and rosmarinic acid	Chitosan/gelatin film	2%	- Improved the antimicrobial activity against *E. coli* - and *L. monocytogenes*. - Improved the antioxidant activity against DPPH and ABTS.	[104]
Angelica EO	Polyphenols such as rosemanol, epirosmanol, and rosmarinic acid	Polylactic acid active film	4%	- Improved the antioxidant activity against DPPH.	[111]
Turmeric EO	Polyphenols such as rosemanol, epirosmanol, and rosmarinic acid	Chitosan film	1.5 μL/cm^2^ and 3.0 μL/cm^2^	- Improved the antimicrobial activity against *Aspergillus flavus* growth and *conidial formation.* - Inhibited aflatoxin biosynthesis.	[118]

## 5. Application of Bioactive Films Incorporated with Essential Oils in Food

Recently, active packaging has received a huge amount of attention for its antioxidant and antimicrobial properties to increase the shelf-life of food products. Several approaches have been introduced by incorporating antioxidant and antimicrobial compounds to improve the bioactivity of biomaterials used for packaging. Essential oils (EOs) and plant extracts are regarded as proper alternatives to artificial food additives in bioactive films for preservation because of their safety and outstanding antioxidant and antimicrobial activities [119]. For example, the antimicrobial activity of chitosan film was significantly increased with the integration of melaleuca essential oil (MEO) and clove essential oil (CEO) [107]. Similarly, the antioxidant activity of pullulan-based film was increased with rockrose essential oil incorporation [106]. Other researchers, like Luís, Pereira [116] and Yuan, Chen [120], also reported the ability of essential oils to protect different types of food against oxidation and pathogenic and spoilage microorganisms. Bioactive films having essential oils were used for the preservation of these food products.

### 5.1. Fruits and Vegetables

Agricultural products, especially fruits and vegetables, are highly perishable postharvest in the natural environment; thus, their preservation is an ongoing problem for the food industry. Deterioration of fruit and vegetables occurs as a result of water loss, high respiration rates, microbial invasion, and oxidation. Therefore, it is crucial to create a suitable system to extend the post-harvest storage period and ensure the quality of fruits and vegetables [121]. Recently, bioactive films with essential oils have received substantial consideration as an efficient, easy, and affordable method to slow down the deterioration of fruits and vegetables [122]. The antimicrobials (essential oils) are embedded in active films, allowing the active biocide compounds to be released from the packaging for a longer period, extending the effects during food transport and storage [123]. Cassava starch-based film supplemented with clove EO showed antifungal activity against *C. musae* and *C. gloeosporioides*. The film preserved the quality and reduced weight loss of banana varieties such as BRS Tropical, Prata-Anã, BRS Conquista, and Grand Nine during storage [124]. PLA and poly (3-hydroxybutyrate-4-hydroxybutyrate) film containing angelica EO demonstrated better antioxidant ability, which effectively retarded the oxidation process by preventing LOX and PPO enzymatic activity in peaches and extended their shelf-life without compromising quality for more than 15 days [123]. Zhou, He [114] developed carboxymethyl chitosan/pullulan composite film supplemented with galangal essential oil, which effectively preserved mango fruits regarding fruit weight loss, firmness, TSS, and TA for 9 days compared to control film at room temperature. The effect of the addition of different polysaccharides, like tragacanth gum, xanthan gum, gum arabic, and pullulan, on the release of thyme EO from chitosan was investigated by Lian, Shi [125] to preserve nectarine fruit. Compared to other polysaccharides, gum arabic delayed thyme EO release and consequently reduced fruit lesions in nectarines compared to other polysaccharide-added films after 60 h of storage. Moreover, Passafiume, Tinebra [126] used neem essential oil to preserve sliced mango for 9 days at 4 °C. Biodegradable multilayer chitosan/starch-based film with added cinnamon EO (0.33 g) reduced weight loss, retained the freshness and firmness of cherry tomatoes, and outperformed the antibacterial and preservative qualities of polyethylene film for two weeks of storage [127]. Previously, methylcellulose and polycaprolactone/alginate-based composite film with essential oil supplementation considerably reduced *L. monocytogenes* and *E. coli* growth in fresh broccoli during short-term storage [128]. Similarly, the antimicrobial activity of gelatin-based film was improved against *E. coli* and *S. aureus* with the integration of banana leaf EO and preserved cherry tomatoes during 14 days of storage time, as reported by Kamari, Halim [129]. The application of thyme EO in sweet potato starch film was investigated for spinach leaves [130]. The numbers of *Salmonella typhimurium* and *Escherichia coli* in fresh spinach leaves were reduced with EO incorporation from the levels that were expected in five days and the shelf-life of spinach was enhanced. Some other fruits and vegetables preserved through bioactive films having essential oils are summarized in Table 2.

### 5.2. Meat and Its Products

Meat and its products are perishable due to the presence of high nutrient content, moisture content, endogenous proteases, and neutral pH, creating ideal conditions for bacterial and biochemical deterioration. The incorporation of bioactive compounds such as essential oils into packaging materials to extend the shelf-life of meat has become an effective method. There have been several reports about the use of active films for meat and meat product preservation. Mung bean protein isolate/pullulan film used for the preservation of minced beef showed improved antimicrobial and antioxidant properties with the addition of marjoram (*Origanum majorana* L.) essential oil (MEO). Compared to the control film, it reduced the bacterial population and chemical properties (TBARS, TVB-N, and pH) of minced beef after 14 days of storage time, indicating protective effects to increase the shelf-life [131]. The supplementation of starch-based edible film with torch ginger (*Etlingera elatior Jack*) inflorescence essential oil (TGIEO) enhanced its antibacterial and antioxidant properties and increased chicken meat shelf-life by presenting a lower coliform count and TBARS value during a storage period of 6 days [132]. Chitosan (CH) blended with MMT-based nanocomposite film extended poultry meat shelf-life by controlling its TBARS, pH, and color [133]. Later that year, in 2019, these researchers created a rosemary EO (REO)-enriched CH/MMT matrix to preserve raw poultry meat [134]. The bioactive film demonstrated lower O_2_ permeability and better antibacterial properties by reducing bacterial populations, lipid oxidation, and discoloration of poultry.

Chitosan-based edible film showed enhanced inhibitory effects against coliform bacteria, TVC, and PTC with the incorporation of Trachyspermum ammi EO, which subsequently extended the shelf-life of meat [135]. Potato starch/apple peel pectin-based composite film containing microencapsulated *Zataria multiflora* essential oil and ZrO_2_ nanoparticles was used for quail meat packaging. The chemical characteristics of quail meat wrapped in active film showed the positive impact of encapsulated essential oil and ZrO_2_ nanoparticles in extending its shelf-life [98]. The supplementation of Cassava starch film with oregano essential oil demonstrated better antimicrobial and antioxidant activities to prevent the oxidation of ground beef [136]. The resulting film preserved the meat against lipid oxidation for 3 days under refrigerated conditions. Likewise, the incorporation of lemongrass EO into cassava starch film also reduced microbial counts during storage [137]. Shen, Zhou [103] prepared *Dioscorea zingiberensis* (DZW) starch film loaded with oregano essential oil (OEO), which presented better antioxidant activity and inhibited the growth of pathogenic bacteria in chicken meat for 7 days stored at 4 °C. Recently, we used forsythia essential oil in ASKG-based film and extended the shelf-life of chicken and lamb meat for 9 and 12 days, respectively, when stored at 4 °C [138]. These findings expressed the effectiveness of OEO-added DZW starch film in maintaining the quality and extending the shelf-life of fresh meat. Some other examples of meat and meat product preservation are given in Table 2.

### 5.3. Fish and Fish Products

Fish and seafood are very important due to their nutritional value, popularity as a delicacy, and health benefits; however, their high moisture content, neutral pH, and large quantities of small molecules make them highly perishable by providing the perfect conditions for microbial and biochemical deterioration [139]. Endogenous enzymatic reactions, microbial activity, and oxidation occur in fish shortly after death, so proper preservation methods should be adopted to preserve quality and increase shelf-life [140]. Recently, natural preservatives have been the focus of fishery product preservation; in this regard, essential oils (EOs) are receiving huge attention owing to their impressive antimicrobial and antioxidant activities [141]. Cao and Song [142] developed Bombacaceae gum film having cinnamon leaf essential oil for fresh salmon fillet preservation. The POV and TBARS values of fish fillets wrapped in active film having essential oil were found to be lower compared to the control sample after 15 days at 4 °C, reflecting a reduction in lipid oxidation to extend the shelf-life of fish fillets. Similarly, starch and oregano essential oil (OEO)-integrated film was developed to prolong frozen fish fillet shelf-life [143]. The addition of sage EO to gelatin, alginate, and chitosan-based film effectively retarded microbial growth, including *Shewanella* spp. *A* and *Pseudomonas* spp., in fish burger stored at 4 °C [144]. Similarly, rice starch film supplemented with oregano essential oil was used to preserve fish fillets. The active film displayed higher resistance against microbial growth and lipid oxidation in fish fillets during a storage time of 6 days [143]. Fish packaged in Cassava starch film enriched with Citrus lemon peel extract demonstrated low total volatile basic nitrogen (TVB-N) and peroxide values compared to that packaged with the control film [145]. Soy protein isolate, montmorillonite, and clove EO-based nanocomposite film was used for bluefin tuna (*Thunnus thynnus*) fillet preservation. The active film reduced lipid oxidation and decreased the final counts of MBC, HSPB, TVC, *Enterobacteriaceae*, and *Pseudomonas* spp. during a storage time of 16 days under refrigerated conditions [146]. Some of the other applications of active films having essential oils are summarized in Table 2.

### 5.4. Dairy Products

Milk and other dairy products, including cream, fermented cheese, yogurt, etc., are good sources of nutrients for growth and health maintenance [91]. However, several extrinsic factors like microorganisms, oxygen, moisture, and light can have detrimental effects on dairy products, including microbial deterioration, oxidation, undesirable odor, and discoloration [147]. The use of active films on dairy products assists in preventing unwanted changes during handling and storage. Many researchers have reported the application of active films in the preservation of dairy products. Seydim, Sarikus-Tutal [90] indicated that whey protein (WPI)-based film having garlic or oregano EO applied to Kasar sliced cheese restricted microbial growth, assuring a 15-day shelf-life of the product. Likewise, cinnamon EO was added to sodium alginate-based film to increase the shelf-life of paneer under refrigerated conditions (4 °C). The film retarded microbial growth and maintained the texture and sensory properties of paneer for 13 days [148]. The applications of active films having EOs in dairy products are summarized in Table 2.

### 5.5. Bread and Bakery Products

Bakery products are mostly consumed by people for breakfast all over the world, but they can be used to replace lunch and dinner. However, many bakery products like bread and cakes at room temperature have a limited lifespan of 3–5 days if no preservatives are used [91]. Bakery products go through several chemical, microbial, and physical changes during their lifespan. The chemical and physical changes result in loss of freshness, crispness, and reduction in taste and texture, while microbial spoilage brings undesirable changes in appearance due to bacteria, mold, and yeast growth [149]. Bioactive films not only prevent microbial growth but also maintain the texture of bakery products [150]. The applications of active films having EOs in bakery products are summarized in Table 2.

### 5.6. Nuts

During storage, nuts are prone to lipid oxidation, which can lead to undesirable taste, and smell, degradation of nutrients, and even produce some toxic chemicals [151]. Studies have demonstrated that active films having antioxidants can prevent lipid oxidation in nuts and increase their shelf-life (Table 2). Furthermore, active films prevent nuts from mechanical damage during stress and transportation, extending their shelf-life [152].

**Table 2 foods-12-04169-t002:** Application of films having EOs for food packaging and preservation.

Film Composition	Essential Oil	Food Application	Findings	Reference
Chitosan/collagen protein	Cinnamon-perilla essential oil	Sea bream fillets	- Delayed lipid oxidation. - Increased antimicrobial activity. - Extended shelf-life for 6–8 days.	[153]
Pectin	Oregano essential oil, ginger essential oil	Yellow croaker	- Protective effect against protein oxidation, and prevention of endogenous enzymatic activity. - Enhanced shelf-life for 7 days during ice storage.	[154]
Chitosan/alginate/gelatin	Sage	Carp fish burger	- Reduced pH, TBA, TVC, PTBC, *Pseudomonas*, and *Shewanella* for 20 days. - Sensory score was increased.	[144]
Soy protein	Clove	Bluefin tuna (*Thunnus thynnus*) fillet	- TBV-N, TBA, TVC, MBC, and HSPB decreased. - Inhibited *Pseudomonas* spp. and *Enterobacteriaceae*. - Preserved food for 17 days.	[146]
Bombacaceae gum	Cinnamon leaf essential oil	Salmon fillets	- Increased antimicrobial activity. - Retarded lipid oxidation, and - malonaldehyde and hydroperoxide generation in salmon for 15 days.	[142]
Chitosan	Ginger	Cobia (Rachycentron canadum) fish steak	- Retarded lipid oxidation. - *Pseudomonas*, *Brochothrix thermosphacta* reduced. - Enhanced sensory score and shelf-life (15 days).	[155]
Қ-Carrageenan	Red cabbage extract (*Brassica oleraceae*)	Rainbow trout fillets	- Helped to monitor the freshness and quality of rainbow trout fillets (*Oncorhynchus mykiss*) by changing the color due to quality deterioration.	[156]
Furcellaran/carboxymethyl cellulose	Lingonberry extract	Salmon (*Salmo salar*) fillets	- Inhibited microbial growth and lowered total bacteria count. - Inhibited the formation of biogenic amines.	[157]
Job’s tears starch	Clove bud essential oil	Pork belly	- Exhibited a lower degree of lipid oxidation determined by peroxides and thiobarbituric acid active substances. - Effectively maintained the freshness and quality of pork belly during storage.	[158]
Chitosan	Thyme essential oil	Beef	- Improved the antimicrobial activity of the film against *E. coli* and *Bacillus subtilis* spp. - Extended the shelf-life to 6 days at 4 °C.	[99]
Chitosan	Oregano essential oil	Chicken fillets	- Reduced the population of food spoilage microorganisms. - Extended the shelf-life of frozen chicken up to 12 days.	[159]
Chitosan	Apricot (*Prunusarmeniaca*) kernel essential oil	Spiced beef	- Exhibited antimicrobial effects against *L. monocytogenes*. - Maintained the texture, color, and taste of the meat stored at 4 °C.	[160]
*Dioscorea zingiberensis* starch	Oregano essential oil	Chicken	- Exhibited antibacterial activity against *B. subtilis*, *E. coli*, and *S. aureus.* - Reduced total bacterial count and preserved chicken meat for 7 days.	[103]
Potato starch (St)/apple peel pectin (Pec)	*Zataria multiflora* essential oil	Quail meat	- Improved antibacterial property by reducing microbial count. - Reduced lipid oxidation and extended shelf-life of meat (12 days).	[98]
Mung bean protein isolate/pullulan	Marjoram essential oil	Minced beef	- Showed antibacterial activity against *Staphylococcus aureus* and *Escherichia coli* and retarded microbial growth. - Reduced lipid oxidation and preserved meat quality for up to 14 days.	[131]
Whey protein	Rosemary oil	Lamb meat	- Significantly reduced microbial growth. - Retarded lipid oxidation and lipolysis and preserved meat for 7 days.	[161]
Corn starch	Zataria multiflora	Ground beef patties	- Improved antibacterial property by reducing microbial count. - Reduced lipid oxidation and extended shelf-life of meat to 20 days.	[162]
Starch	Torch ginger essential oil	Chicken meat	- Improved antibacterial and antioxidant properties (lower coliform count and TBARS value). - Enhanced storage life of chilled meat up to 6 days.	[132]
Whey protein	Rosemary essential oil	Lamb meat	- Film significantly reduced the bacterial counts of treatment groups. - Increased the shelf-life of meat (15 days) compared to the control meat (6 days).	[163]
Cassava starch	Clove EO	Bananas	- Showed antifungal activity against *C. musae* and *C. gloeosporioides*. - Maintained quality attributes and reduced water loss.	[124]
PLA	Angelica EO	Peaches	- Exhibited better antioxidant capacity. - Inhibited LOX and PPO enzymatic activity and preserved peach fruit for more than 15 days.	[123]
Carboxymethyl chitosan/pullulan	Galangal essential oil	Mangoes	- Reduced fruit weight loss. - Preserved fruit regarding firmness, titratable acidity, and soluble solids for 15 days.	[114]
Chitosan	Thyme essential oil	Nectarine fruit	- Exhibited good antifungal effect on nectarines. - Lowed fruit lesions in nectarines after 60 h of storage.	[125]
Alginate with apple puree	lemongrass	Apples	- Exhibited antimicrobial activity against *L. innocua.* - Maintained the firmness and sensory quality for 21 days.	[164]
PLA	Ginger essential oil, Angelica essential oil	Peaches	- Inhibited the increase in MDA content and PPO and LOX activities. - Reduced moisture loss. - Preserved color and quality of peaches for 30 days.	[111]
Chitosan/Casein	Oregano essential oil	Cherry tomatoes	- Reduced weight loss, shrinkage, and titratable acidity. - Inhibited fungal growth for 28 days at 4 °C.	[165]
Poly(lactic acid)/poly(ε-caprolactone)	Thymol	Hot peppers	- The active film could maintain the quality of fresh hot peppers and extend the post-harvest life.	[166]
Chitosan/starch	Cinnamon leaf essential oil	Tomatoes	- Retained the freshness and firmness and reduced weight loss. - Improved antibacterial activity and preserved cherry tomatoes for 2 weeks.	[127]
Methylcellulose	Rosemary extract/Asian spice essential oil	Broccoli	- Controlled *L. monocytogenes* and *E. coli* growth. - Preserved broccoli for 12 days at 4 °C.	[128]
Gelatin	Banana leaf EO	Cherry tomatoes	- Lowered weight loss and browning index values. - Preserved cherry tomatoes for 14 days.	[129]
Potato starch	Thyme EO	Spinach	- Reduced the number of *Escherichia coli* and *Salmonella typhimurium*. - Extended the shelf-life for 5 days.	[130]
Whey protein	Garlic/oregano EO	Kasar cheese	- Restricted microbial growth. - Extended the shelf-life up to 15 days.	[90]
Sodium alginate	Cinnamon EO	Paneer	- Retarded microbial growth. - Maintained texture and sensory properties. - Extended the shelf-life from 5 days to 13 days.	[148]
Zein	*Rosmarinus officinalis* essential oil	Cheese	- The film showed significant antimicrobial activity against *L. monocytogenes*, *S. aureus,* and aerobic mesophilic bacteria when applied to cheese slices.	[167]
κ-Carrageenan	Black carob extract	Cheese	- The novel antioxidant film slowed down the oxidation of the cheese.	[168]
Gelatin/chitosan	Boldo extract	Sliced Prato cheese	- At 4 °C, the film exhibited a protective effect against lipid oxidation and inhibited microorganism growth on sliced Prato cheese.	[169]
Carboxymethyl cellulose (CMC)-polyvinyl alcohol (PVA)	Cinnamon essential oil	Bread	- The films containing 1.5 and 3% CEO were highly effective against P. digitatum, increasing the shelf-life of bread.	[170]
Poly (lactic acid)/poly (butylene-succinate-co-adipate)	Thymol	Bread	- The film could extend the shelf-life of bread to 9 days.	[171]
Chitosan/Poly(ε-caprolactone) (PCL)	Grapefruit seed extract (GFSE)	Bread	- Inhibited the growth of *Escherichia coli* and *Pseudomonas aeruginosa*. - No mold growth was observed on the bread packaged with film containing ≥ 1.0 mL/g GFSE after 7 days.	[172]
Cashew gum/gelatin	Cymbopogon citratus essential oil	Bread	- The film preserved bread for 6 days compared to 3 days for commercial packaging.	[173]
Starch/gum	Grapefruit seed extract	Rice cakes	- The film inhibited the growth of *B. cereus* and *P. citrinum* during rice cake storage.	[174]
Chitosan	Mango leaf extract	Cashew nuts	- The film significantly reduced oxidation and preserved cashew nuts for 28 days.	[175]
Chitosan	Green tea extract	Fresh walnut kernels	- The film significantly inhibited lipid oxidation and fungal growth in fresh walnut kernels.	[176]
Indian gooseberry puree/methylcellulose	Indian gooseberry extract	Cashew nuts	- The film enhanced the shelf-life of cashew nuts during storage.	[177]

## 6. Conclusions

Essential oils can be used as natural food additives. The impact of essential oils on bioactive films and their use in food preservation, focusing on their antioxidant and antibacterial properties, is an important research area. Because essential oils contain bioactive chemicals such as terpenes, phenolic compounds, and volatile compounds, they may be used as natural antibacterial and antioxidant agents in food preservation. Essential oils have been combined with bioactive films to protect food from oxidation and microbiological growth, making them a sustainable food preservation technology. It is necessary to investigate their additional functions, as well as their significance in bioactive films and preservation of food through their antioxidant and antibacterial abilities. In the same way, innovative approaches for reducing essential oil odor can improve culinary applications. Systems for encapsulation and release are potential choices. This article also highlights the antibacterial efficacy of essential oil-containing films for food preservation and examines their manufacturing techniques. These ingredients may be an appealing delivery option for increasing phenolic stability in foods and the shelf-life of edible food items. Furthermore, this paper offers an overview of current knowledge on essential oil extraction, their effects on bioactive films and food preservation, in addition to the advantages and disadvantages of implementing them to preserve food products.

## Figures and Tables

**Figure 1 foods-12-04169-f001:**
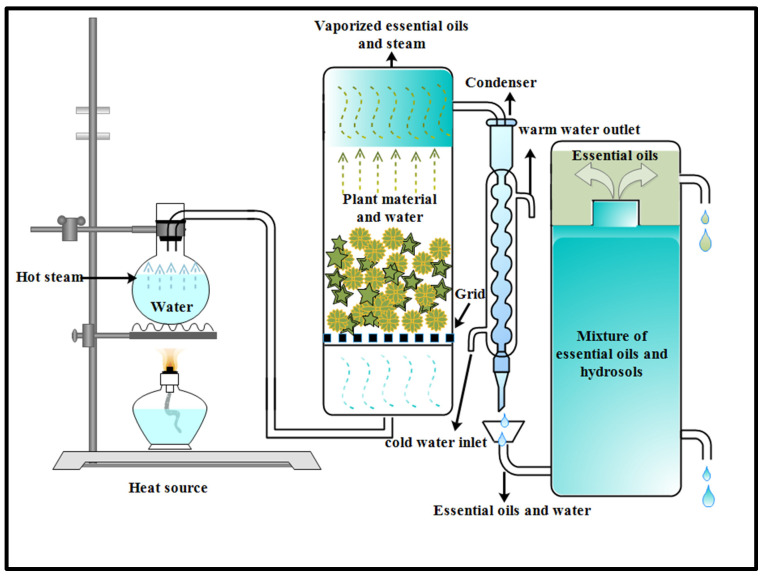
Diagram illustrating essential oil extraction by the steam distillation method.

**Figure 2 foods-12-04169-f002:**
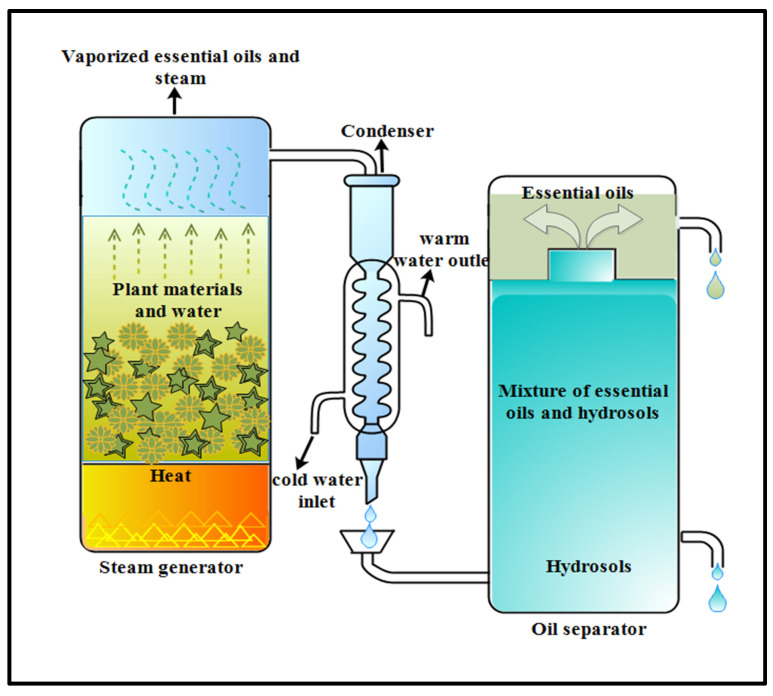
Diagram illustrating essential oil extraction by the hydrodistillation method.

**Figure 3 foods-12-04169-f003:**
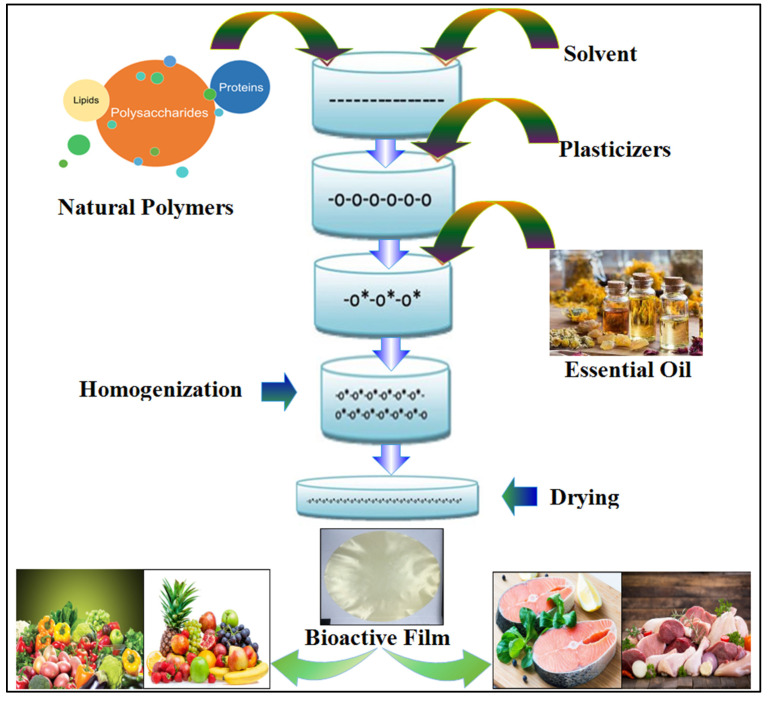
Preparation of edible films with tailored properties for food preservation.

## Data Availability

Data is contained within the article.

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
