# Peer review of "The Extraction and Impact of Essential Oils on Bioactive Films and Food Preservation, with Emphasis on Antioxidant and Antibacterial Activities—A Review"

_foods, 2023, doi:10.3390/foods12224169_

Round 1
Reviewer 1 Report
Comments and Suggestions for Authors
The manuscript entitled 'The extraction and impact of essential oils on bioactive films and food preservation, with emphasis on antioxidant and anti-bacterial activities - A review' overall contains interesting scientific information. However, I feel that there are some gaps concerning the comparison of the numerical data of the different essential oils mentioned. In addition, it would be good, given the food area, for the percentages of essential oils that can be used and approved by the FDA to be included.
The introduction appears to be relevant and consistent with the proposed title. However, its development could be increased on the basis of further research and insights into the addition of essential oils within edible coatings or formulations of nanoemulsions and microencapsulation for the creation of bio-based packaging.
Below I provide some citations that may be useful within this review.
Culmone, A., Mirabile, G., Tinebra, I., Michelozzi, M., Carrubba, A., Bellardi, M. G., ... & Torta, L. (2023). Hydrolate and EO Application to Reduce Decay of Carica papaya during Storage. Horticulturae, 9(2), 204.
Line 36. Delete the space after the full stop.
Paragraph 2.2 Distillation:
The methods described for distillation are well defined, however, it is correct to specify that essential oil, to be so defined, is derived solely and exclusively from a physical process, whereas all other processes described result in similar, not the same, products.
line 235. Figure 3. The images below are distorted, the dimensions could be adapted.
line 261. Missing dot in L.; delete a space after WVP;
line 337. Delete space after dot;
line 355. Delete space after full stop;
line 362. Delete space after citation 97;
line 367. You could take a cue from the manuscript 'Passafiume, R., Tinebra, I., Gaglio, R., Settanni, L., Sortino, G., Allegra, A., & Farina, V. (2022). Fresh-Cut Mangoes: How to Increase Shelf Life by Using Neem Oil Edible Coating. Coatings, 12(5), 664.'
line 634 - 635. "In the same way, innovative approaches for reducing essential oil odour can improve culinary applications. Systems for encapsulation and release are potential choices'. This statement, although interesting, was not elaborated upon enough in the manuscript, e.g. it could be expanded upon by including publications containing sensory analyses of foods treated with essential oils.
Check for double spaces within the text.
Author Response
Comment # 1 The manuscript entitled 'The extraction and impact of essential oils on bioactive films and food preservation, with emphasis on antioxidant and anti-bacterial activities - A review' overall contains interesting scientific information. However, I feel that there are some gaps concerning the comparison of the numerical data of the different essential oils mentioned. In addition, it would be good, given the food area, for the percentages of essential oils that can be used and approved by the FDA to be included.
Response # 1 Thank you so much for your hard work and valuable suggestions. The percentage of some essential oils used in bioactive films is given in the “Bioactivity of essential oils-added film” section. However, the concentrations of different essential oils are different from each other to give particular bioactive properties, which is mentioned in line 458 and 459, and suggested that this area needs further research in the future to optimize the level of essential oils.
Comment # 2 The introduction appears to be relevant and consistent with the proposed title. However, its development could be increased based on further research and insights into the addition of essential oils within edible coatings or formulations of nanoemulsions and microencapsulation for the creation of bio-based packaging.
Below I provide some citations that may be useful within this review.
Culmone, A., Mirabile, G., Tinebra, I., Michelozzi, M., Carrubba, A., Bellardi, M. G., ... & Torta, L. (2023). Hydrolate and EO Application to Reduce Decay of Carica papaya during Storage. Horticulturae, 9(2), 204.
Response # 2 We appreciate your suggestion. The reference was very helpful and we cited it in a suitable section (line 62).
Comment # 3 Line 36. Delete the space after the full stop.
Response # 3 Done as suggested.
Comment # 4 Paragraph 2.2 Distillation:
The methods described for distillation are well defined, however, it is correct to specify that essential oil, to be so defined, is derived solely and exclusively from a physical process, whereas all other processes described result in similar, not the same, products.
Response # 4 Done as suggested.
Comment # 5 line 229. Figure 3. The images below are distorted, the dimensions could be adapted.
Response # 5 The figure 3 was revised as suggested.
Comment # 6 line 254. Missing dot in L.; delete a space after WVP;
Response # 6 Done as suggested.
Comment # 7 line 320. Delete space after dot;

Reviewer 2 Report
Comments and Suggestions for Authors
The review article entitled 'The extraction and impact of essential oils on bioactive films and food preservation, with emphasis on antioxidant and antibacterial activities - A review', provides a literature review on the extraction of essential oils and their influence on bioactive films and food preservation. Additionally, the article highlights the advantages and disadvantages of using them to preserve food products, as well as the benefits of their application. The article uses clear, concise language, and technical term abbreviations are explained upon their first use. Additionally, the article outlines the ability of essential oils to support the function of bioactive films, thus preserving food due to their antioxidant and antimicrobial properties. The antimicrobial effectiveness of composite films incorporating essential oils is also emphasised. Nevertheless, details concerning the production techniques of these films were only broadly introduced.
I read the article on extraction methods for natural essential oils with great interest. It provided thorough information on the advantages and disadvantages of the different methods. While I appreciate that it is challenging to cover all aspects in one article, I have some constructive feedback to share.
1. The literature appears to be largely up to date with 46 out of 174 items featuring scientific papers published after 2020. I did, however, only come across one paper from 2023. It is quite remarkable how rapidly science is evolving, with 2023 already upon us. Consequently, it would be advisable to scrutinise the latest papers in this field.
2. More emphasis ought to be given to distillation under reduced pressure and vacuum, due to the potential of conducting these procedures at lower temperatures.
3. Limited research has been conducted on the use of ultrasound in the extraction process, but it is believed to enhance extraction efficiency.
4. The authors exclusively explore the bactericidal and fungicidal properties of composites that comprise essential oils/extracts. Nevertheless, it is worth noting that these materials often possess sensory properties and can function as sensors to detect alterations in packaging, such as those found in food products.
There are also editorial errors in the work, for example:
Line 65 life.[15]. Should be life [15].
Line 86. similarly, "modest critical conditions. [25]." Please check AND remove full stops before quotations.
Line 91-92. " including temperature, pretreatment of materials, temperature, particle size, time, pressure, solvent-to-feed ratio, and solvent flow rate...". Temperature occurs twice.
Line 97: correct CO2 formula (lower index)
Line 208-209: correct citation form "and allergies and alter the immune system (Ferhat and others 2007a). "
In conclusion I think the paper is quite interesting and suitable for publication in Foods with minor corrections.
Author Response
Reviewer 2 comments and responses
The review article entitled 'The extraction and impact of essential oils on bioactive films and food preservation, with emphasis on antioxidant and antibacterial activities - A review', provides a literature review on the extraction of essential oils and their influence on bioactive films and food preservation. Additionally, the article highlights the advantages and disadvantages of using them to preserve food products, as well as the benefits of their application. The article uses clear, concise language, and technical term abbreviations are explained upon their first use. Additionally, the article outlines the ability of essential oils to support the function of bioactive films, thus preserving food due to their antioxidant and antimicrobial properties. The antimicrobial effectiveness of composite films incorporating essential oils is also emphasised. Nevertheless, details concerning the production techniques of these films were only broadly introduced.
I read the article on extraction methods for natural essential oils with great interest. It provided thorough information on the advantages and disadvantages of the different methods. While I appreciate that it is challenging to cover all aspects in one article, I have some constructive feedback to share.
- The literature appears to be largely up to date with 46 out of 174 items featuring scientific papers published after 2020. I did, however, only come across one paper from 2023. It is quite remarkable how rapidly science is evolving, with 2023 already upon us. Consequently, it would be advisable to scrutinise the latest papers in this field.
- More emphasis ought to be given to distillation under reduced pressure and vacuum, due to the potential of conducting these procedures at lower temperatures.
- Limited research has been conducted on the use of ultrasound in the extraction process, but it is believed to enhance extraction efficiency.
- The authors exclusively explore the bactericidal and fungicidal properties of composites that comprise essential oils/extracts. Nevertheless, it is worth noting that these materials often possess sensory properties and can function as sensors to detect alterations in packaging, such as those found in food products.
There are also editorial errors in the work, for example:
Comment # 1 Line 63 life.[15]. Should be life [15].
Response # 1 Done as suggested
Comment # 2 Line 192. similarly, "modest critical conditions. [26]." Please check AND remove full stops before quotations.
Response # 2 Done as suggested
Comment # 3 Line 197-198. " Including temperature, pretreatment of materials, temperature, particle size, time, pressure, solvent-to-feed ratio, and solvent flow rate...". Temperature occurs twice.
Response # 3 The extra “temperature “word was removed.
Comment # 4 Line 202: correct CO2 formula (lower index).
Response # 4 Corrected according to the suggestion.
Comment # 5 Line 183-184: correct citation form "and allergies and alter the immune system (Ferhat and others 2007a). "
Response # 5 The citation was corrected accordingly.
In conclusion I think the paper is quite interesting and suitable for publication in Foods with minor corrections.

Reviewer 3 Report
Comments and Suggestions for Authors
The review article compiles the results of “extraction and impact of essential oils on bioactive films and food preservation, with emphasis on antioxidant and anti-bacterial activities. The information (source, extraction method, and food packaging application of essential oil) provided in the manuscript is not novel and well reported in earlier studies. However, the review presented some interesting information but should be updated with additional information.
Some critical comments
1. Table 1; what do authors mean by “essential oil type” and then why is only oregano highlighted in three places while not others?
2. Further, update Table 1 with essential oils source (botanical name of the plant and parts).
3. This review article completely ignores the importance of some essential oils and their composite properties. For example, tea tree oil, lavender, etc. Please update the information by referring to the following articles.https://www.sciencedirect.com/science/article/pii/S0300944020312212, https://doi.org/10.1080/10412905.2016.1232665.
4. Table 2 is updated with the physical properties of the film. The author should mention “essential oil” instead of essential oil type.
5. Grape seed extract or grape seed oil?
6. The review article should be updated with colorful images in the application part.
Comments on the Quality of English LanguageMinor editing of English language required
Author Response
Reviewer 3 comments and responses
Comment # 1 Table 1; what do authors mean by “essential oil type” and then why is only oregano highlighted in three places while not others?
Response # 1 The word “type” in table 1 was a mistake so we removed it from the manuscript, while the oregano is written 3 time because it was used by different researchers in the development of different types of films.
Comment # 2 Further, update Table 1 with essential oils source (botanical name of the plant and parts).
Response # 2 Thank you so much for your valuable suggestion, we did not mention the plant or part from which the oil was extracted because most of the articles did not give the name of plant or part from which the oils were extracted.
Comment # 3 This review article completely ignores the importance of some essential oils and their composite properties. For example, tea tree oil, lavender, etc. Please update the information by referring to the following articles.https://www.sciencedirect.com/science/article/pii/S0300944020312212, https://doi.org/10.1080/10412905.2016.1232665.
Response # 3 Thank you so much for your suggestion. The reference was very helpful and we included about the “tea tree oil” in the manuscript (line 396 and 397).
Comment # 4 Table 2 is updated with the physical properties of the film. The author should mention “essential oil” instead of essential oil type.
Response # 4 The word “type” in the table 2 wasn’t appropriate, so corrected it by removing it from the manuscript.
Comment # 5 Grape seed extract or grape seed oil?
Response # 5 The word in the manuscript is grape seed extract.
Comment # 6 The review article should be updated with colourful images in the application part.
Response # 6 The figure in the applicated was revised as suggested.

Round 2
Reviewer 3 Report
Comments and Suggestions for Authors
-
Comments on the Quality of English LanguageMinor editing of English language required